# Respiratory symptoms after coalmine fire and pandemic: A longitudinal analysis of the Hazelwood Health Study adult cohort

Tyler J. Lane[1]*, Matthew Carroll[2], Brigitte M. Borg[1,3], Tracy A. McCaffrey[4], Catherine L. Smith[1], Caroline X. Gao[1,5], David Brown[1], Amanda Johnson[1], David Poland[2], Shantelle Allgood[2], Jillian Ikin[1], Michael J. Abramson[1]

**1** School of Public Health and Preventive Medicine, Monash University, Melbourne, VIC, Australia, **2** Monash Rural Health Churchill, Monash University, Churchill, VIC, Australia, **3** Respiratory Medicine, The Alfred, Melbourne, VIC, Australia, **4** Department of Nutrition, Dietetics and Food, Monash University, Melbourne, VIC, Australia, **5** Orygen, Centre for Youth Mental Health, The University of Melbourne, Parkville, VIC, Australia

* tyler.lane@monash.edu

## Abstract

The aim of this study was to determine whether the effects of extreme but discrete $PM_{2.5}$ exposure from a coal mine fire on respiratory symptoms abated, persisted, or worsened over time, and whether they were exacerbated by COVID-19. We analysed longitudinal survey data from a cohort residing near a 2014 coalmine fire in regional Australia. A 2016/2017 survey included 4,056 participants, of whom 612 were followed-up in 2022. Items included respiratory symptoms, history of COVID-19, and time-location diaries from the mine fire period, which were combined with geospatial and temporal models of fire-related $PM_{2.5}$. Longitudinal effects of fire-related $PM_{2.5}$ were examined using a mixed-effects logistic regression model. Exacerbation due to COVID-19 was examined using a logistic regression model. $PM_{2.5}$ exposure was associated with chronic cough and possibly current wheeze, chest tightness, and current nasal symptoms 2–3 years post-fire, and chronic cough and current wheeze 8.5–9 years post-fire. Further, the association between $PM_{2.5}$ and chronic cough and possibly current wheeze appeared to increase between the survey periods. While there were no detectable interactions between $PM_{2.5}$ and COVID-19, $PM_{2.5}$ exposure was associated with additional respiratory symptoms among participants who reported a history of COVID-19. In summary, medium-duration exposure to extreme levels of fire-related $PM_{2.5}$ may have increased the long-term risk of chronic cough and current wheeze. While the COVID-19 pandemic started several years after the mine fire, contracting this illness may have exacerbated the effect of fire-related $PM_{2.5}$ through development of additional respiratory symptoms.

## 1 Introduction

Exposure to fine particulate matter < 2.5 μm ($PM_{2.5}$) worsens respiratory health [1,2]. When the source is an extreme combustion event like wildfire, $PM_{2.5}$ is likely even more harmful [3]. In early 2014, a coalmine fire near the Hazelwood power station in regional Victoria,

**Data availability statement:** As per the terms of the contract with the project funder, the Victorian Department of Health, we are unable to make our data public as they contain sensitive and potentially identifiable information. Requests for access may be sent to contact@ hazelwoodhealthstudy.org.au.

**Funding:** This study was funded by the Department of Health, State Government of Victoria (providing salary support for TJL, MC, CLS, CXG, DB, DP, SA, JI, MJA; all others – BMB, TAM, AJ – donated their time), but the manuscript represents the views of the authors and not the Department. The funder (Department of Health) had no role in study design, data collection, analysis or preparation of the manuscript. A draft of this paper was submitted to the funder for comment prior to submission for publication.

**Competing interests:** I have read the journal's policy and the authors of this manuscript have the following competing interests: MJA holds investigator-initiated grants from Pfizer, Boehringer-Ingelheim, Sanofi and GlaxoSmithKline for unrelated research. He has undertaken an unrelated consultancy for Sanofi and received a speaker's fee from GSK. The other authors declare no other competing interests. The remaining authors have declared that no competing interests exist.

Australia, shrouded the adjacent town of Morwell in smoke and ash for six weeks. Daily mean $PM_{2.5}$ reached 1,022 $\mu g/m^3$ in residential areas closest to the mine [4], nearly seven times Environment Protection Authority Victoria's threshold of 150 $\mu g/m^3$ for "extremely poor" air quality [5]. Those residing in Morwell reported numerous symptoms during the coalmine fire including respiratory problems like shortness of breath, chest tightness, and cough [6].

The Hazelwood Health Study was established to investigate how smoke from the coalmine fire affected long-term health. In the years since, $PM_{2.5}$ exposure has been consistently linked to poorer respiratory health. During the fire, $PM_{2.5}$ exposure was associated with increased use of respiratory medical services including general practitioner (GP) and specialist visits [7], hospital admissions, emergency presentations, and ambulance attendances [8,9], and dispensing of inhaled medicines [10]. There is evidence that the fire's effects persisted for at least a few years, with fire-related $PM_{2.5}$ being associated with higher risk of respiratory symptoms measured between 2–4 years post-fire [11,12], accelerated lung ageing and features of chronic obstructive pulmonary disease (COPD) at 4 years [12–14], increased ambulance attendances for respiratory conditions up to 3.5 years post-fire [15], and increased emergency department presentations for respiratory conditions when measured up to 5 [16] and 8 years [17] post-fire. However, there is little evidence of longer-term effects on hospital admissions and ambulance attendances for respiratory conditions [17]. Recent findings suggest some recovery in lung function 7.5 years after the fire [18], though the long-term effects on respiratory symptoms remain unclear.

Late 2019 saw the emergence of another threat to respiratory health, the COVID-19 pandemic. Both particulate matter and COVID-19 can increase inflammatory immune responses in the lungs [2,19]. $PM_{2.5}$ exposures may even increase the likelihood that SARS-CoV-2 infection leads to a cytokine storm [20], or sustained inflammation that can damage the respiratory system [19]. The combination of ambient $PM_{2.5}$ exposure and COVID-19 has been associated with more severe disease [21]. However, it was unknown whether the effects of discrete but extreme $PM_{2.5}$ exposure can be exacerbated by COVID-19 when the fire event occurred several years prior [22].

In this analysis, we addressed the following research questions:

1. Does coalmine fire-related $PM_{2.5}$ affect long-term respiratory symptom trajectory?

2. Does COVID-19 infection worsen the long-term effects of coalmine fire-related $PM_{2.5}$ exposure on respiratory symptoms?

## 2 Methods

This study was pre-registered on the Open Science Framework on 19 July 2022, the month before the follow-up survey commenced, and updated during the survey with a more specific analysis plan on 22 November 2022 [23].

### 2.1 Participants

This analysis uses data from an initial and follow-up survey of the Hazelwood Health Study's Adult Cohort. The cohort was established in the initial survey, administered from 12 May 2016 to 14 February 2017. Using electoral rolls, we identified individuals who were residing in two areas during the coalmine fire (see Fig 1) and invited them to participate in a health survey: Morwell, a town adjacent to the coalmine and whose residents experienced the most smoke exposure, and selected areas with similar socioeconomic characteristics in nearby (~60 km) but unexposed Sale to serve as a control. This resulted in a cohort of 4,056 [24]. For the follow-up survey, from August to December 2022 we sent emails and mobile phone

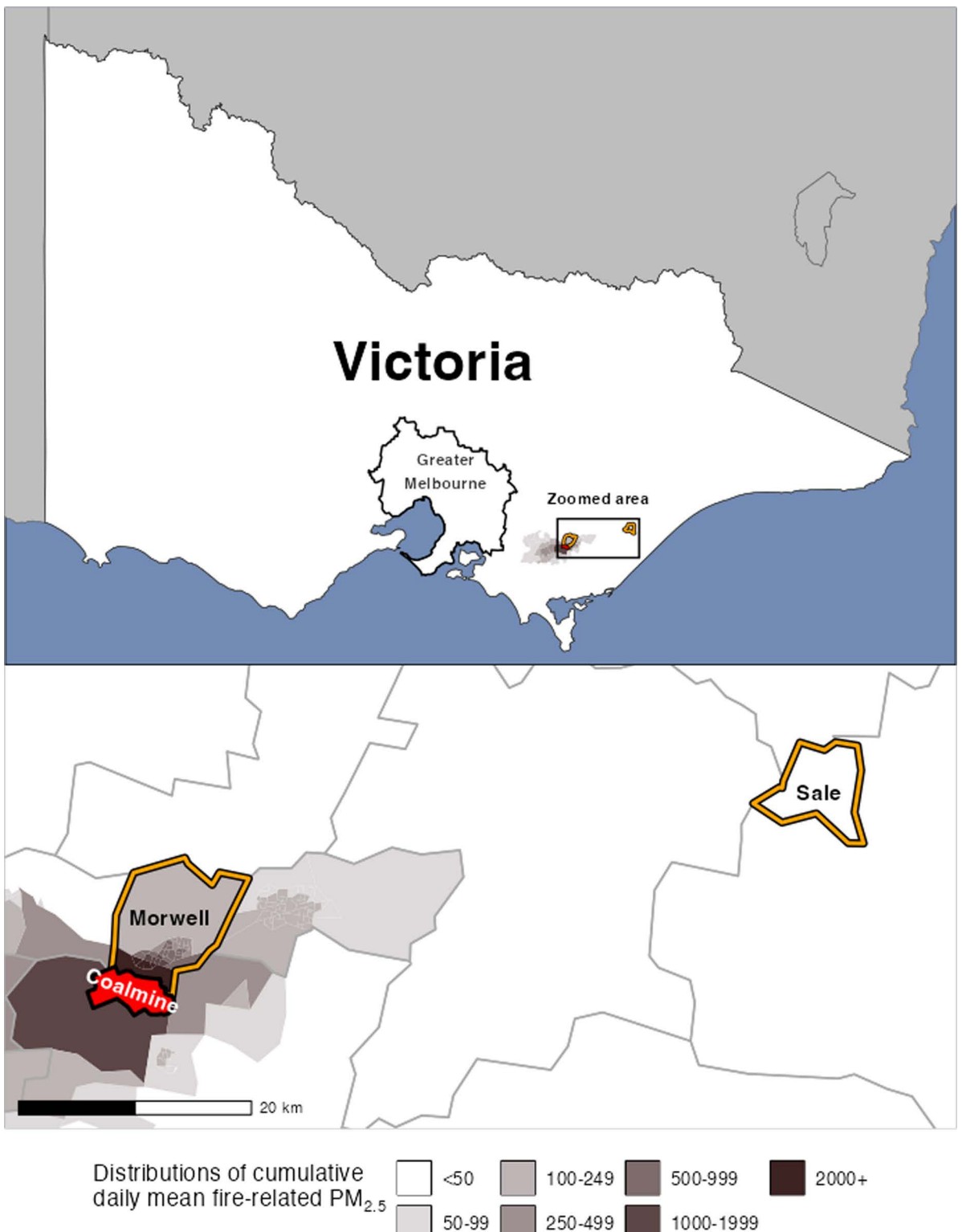

**Fig 1. Maps of Victoria illustrating location of the coalmine and distributions of fire related-PM$_{2.5}$ (February to March 2014) in the surrounding areas, highlighting exposure site Morwell and control site Sale; boundary data from the Australian Bureau of Statistics (source:** https://www.abs.gov.au/AUSSTATS/abs@.nsf/DetailsPage/1270.0.55.006July2011**; copyright information:** https://www.abs.gov.au/website-privacy-copyright-and-disclaimer#copyright-and-creative-commons**) [**45**], coalmine boundary data from the Department of Primary Industries [**46**] and Department of Energy, Environment and Climate Action's DataShare platform (source:** https://datashare.

maps.vic.gov.au/search?q=victorian%20coal%20fields; **copyright information:** https://www.deeca.vic.gov.au/copyright) [47]**, and modelled PM$_{2.5}$ data from Luhar et al. 2020 [4].**

invites to cohort members if they 1) had provided email/mobile numbers, 2) were not part of a separate follow-up survey on mental health outcomes, 3) consented to further contact to participate in a follow-up study, and 4) were not known to be dead. The initial survey was conducted by Computer-Assisted Telephone Interview (CATI), Computer-Assisted Web Interview (CAWI, provided by the Hunter Research Foundation), or paper questionnaire, while the follow-up was conducted primarily by CAWI using the online platform REDCap [25,26]; some follow-up participants received telephone assistance to complete the survey from research staff.

**2.1.1 Exposures.** Participant's daily mean PM$_{2.5}$ exposure during the coalmine fire was determined by blending self-reported time-location diaries, which placed individuals throughout the mine fire period at 12-hour intervals, with modelled estimates of hourly coalmine fire-related PM$_{2.5}$ concentrations [24]. This strategy accounted for individual-level differences in fire-related PM$_{2.5}$ exposure due to participant movement during the mine fire as well as variations in smoke distribution over space and time. PM$_{2.5}$ was both centred at and evaluated in units of 10 μg/m³.

Estimated PM$_{2.5}$ concentrations were generated using chemical transport models, incorporating air monitoring data, coal combustion, and weather conditions. Spatial models achieved a resolution of 100 m² in the area surrounding the mine fire (10.1 km² area), with precision decreasing along with distance from the fire. For more detail, refer to Luhar et al. 2020 [4], particularly Fig 8. Cumulative daily mean fire-related PM$_{2.5}$ distributions are illustrated in Fig 1, showing the locations of the coalmine, exposure site Morwell, and control site Sale. There was little evidence that levels of background ambient PM$_{2.5}$ meaningfully varied between these areas [22]. History of COVID-19 was self-reported using a standardised questionnaire [27], which can be found in S1 Text.

**2.1.2 Outcomes.** Respiratory symptom questions were derived from a modified version of the European Community Respiratory Health Survey (ECRHS) Short Screening Questionnaire and included: current wheeze, chest tightness, nocturnal shortness of breath, resting shortness of breath, current nasal symptoms, chronic cough, and chronic phlegm [28]. Using chronic cough and chronic phlegm responses, we created two additional outcomes: chronic wet cough (chronic cough with phlegm) and chronic dry cough (chronic cough without phlegm). The questions defining each outcome are available in S1 Text.

**2.1.3 Confounders.** We adjusted for potential confounders as determined by a Directed Acyclic Graph created in *DAGitty* [29], focusing on the simplest model that adequately accounted for bias (see Figs A and B in S1 Text). The longitudinal model included demographics (age at the initial [2016/17] survey, transformed with a natural spline with 3 degrees of freedom to account for non-linear effects, and sex), socioeconomic status (Index of Relative Socioeconomic Advantage and Disadvantage (IRSAD) score for residential area at Statistical Area Level 2 based on 2016 census data [30] and educational attainment), and study site. The COVID-19 moderator analysis adjusted for demographics, socioeconomic status, and study site, plus tobacco use (smoker status [current, former, never] and cigarette pack-years [the number of packs of 20 cigarettes smoked per day multiplied years smoked, square root-transformed to account for extreme right skew]), occupational exposure (≥6 months of working in the coalmine or power plant, or ≥6 months working another job with exposure to dust, fumes, smoke, gas, vapour, or mist), and pre-fire diagnoses of asthma and COPD.

## 2.2 Statistical analyses

Descriptive statistics were used to characterise the sample by survey round and area of residence during the coalmine fire. Continuous variables are summarised with medians and interquartile ranges and categorical/dichotomous variables with counts and percentages.

To estimate the longitudinal effects of fire-related $PM_{2.5}$ exposure on risk of respiratory symptoms, we used mixed-effects logistic regressions, which included an interaction term between $PM_{2.5}$ exposure and survey round (initial, follow-up) and a random intercept for participants. Results are interpreted as the effect of a 10 μg/m³ increase in $PM_{2.5}$ exposure on the risk of each respiratory symptom. Longitudinal analyses included all cohort members. To estimate whether COVID-19 moderated the effect of $PM_{2.5}$ on respiratory symptoms, we used logistic regressions and included an interaction term between $PM_{2.5}$ and COVID-19. Results are interpreted as the moderating effect of COVID-19 on the association between fire-related $PM_{2.5}$ exposure and risk of each respiratory symptom. The COVID-19 moderation regression analysis only included cohort members who participated in the follow-up survey.

For each outcome, we first conducted crude analyses, then added all confounders but study site, then added study site, as this may have accounted for residual confounding due to differences in lifestyle, socioeconomic status, and access to health services between Morwell and Sale. To account for missing data (see Table A in S1 Text), we used a random forest multiple imputation [31], with the number of imputations equivalent to the proportion of cases with missing data. Regression results for each imputed dataset were pooled according to Rubin's rules [32]. While survey data are confidential and cannot be shared, we have archived cleaning and analytical code on a public repository [33]. Data were analysed in R [34] using RStudio [35]. More detail about statistical packages is available in S1 Text.

## 2.3 Ethics

This study received approval from the Monash University Human Research Ethics Committee (MUHREC) as part of the Hazelwood Adult Survey & Health Record Linkage Study (Project ID: 25680; previously CF15/872 – 2015000389 and 6066). Cohort members gave written informed consent for each survey in which they participated.

# 3 Results

## 3.1 Descriptives

Descriptive statistics by survey round and residence can be found in Table A in S1 Text. Among Morwell respondents, the prevalence of all respiratory symptoms increased significantly between surveys, as did current wheeze, chest tightness, resting shortness of breath, and current nasal symptoms among Sale respondents (see Fig C in S1 Text).

From the 4,056 members of the original cohort, 2,458 (61%) were invited to participate in the follow-up survey, of whom 612 (25%) participated. National Death Index data [36], which were linked to cohort data after the survey was completed, indicated 143 invited cohort members (5.8%) died prior to the close of the survey window (December 2022). Of the remaining 1,703 non-participants, 1,359 (80%) did not respond to invites, 224 (13%) refused, 68 (4.0%) gave consent but did not complete the survey, and 52 (3.1%) were not contactable. As 8.5% of cohort members ($n = 351$) had missing data, we generated 9 imputations for regression analysis.

Compared to cohort members who did not take part in the follow-up survey, those who did ($n = 612$) were more likely to come from Sale, have lower socioeconomic indicators, be younger, and to be former smokers rather than current smokers. There were few differences

in respiratory symptom outcomes, though follow-up participants were more likely to report nasal symptoms but less likely to report chronic phlegm.

## 3.2 Longitudinal effects of coalmine fire-related PM$_{2.5}$ on respiratory symptoms

Model results on the longitudinal effects of fire-related PM$_{2.5}$ on respiratory symptoms are illustrated in Fig 2 and summarised in Table C in S1 Text. At the initial survey (2–3 years post-fire), crude and adjusted analyses found PM$_{2.5}$ exposure was associated with higher risk of chronic cough. The risk of current wheeze, chest tightness, and current nasal symptoms were also elevated in crude and adjusted models, though each attenuated in magnitude and significance with adjustment for study site (Morwell or Sale).

Between the initial 2016/17 survey and 2022 follow-up (8.5–9 years post-fire), the risk of chronic cough increased in conjunction with PM$_{2.5}$ exposure. Crude models also found current wheeze increased in association with PM$_{2.5}$ exposure between the initial and follow-up survey; while point estimates and precision were similar in adjusted models, there was a slight

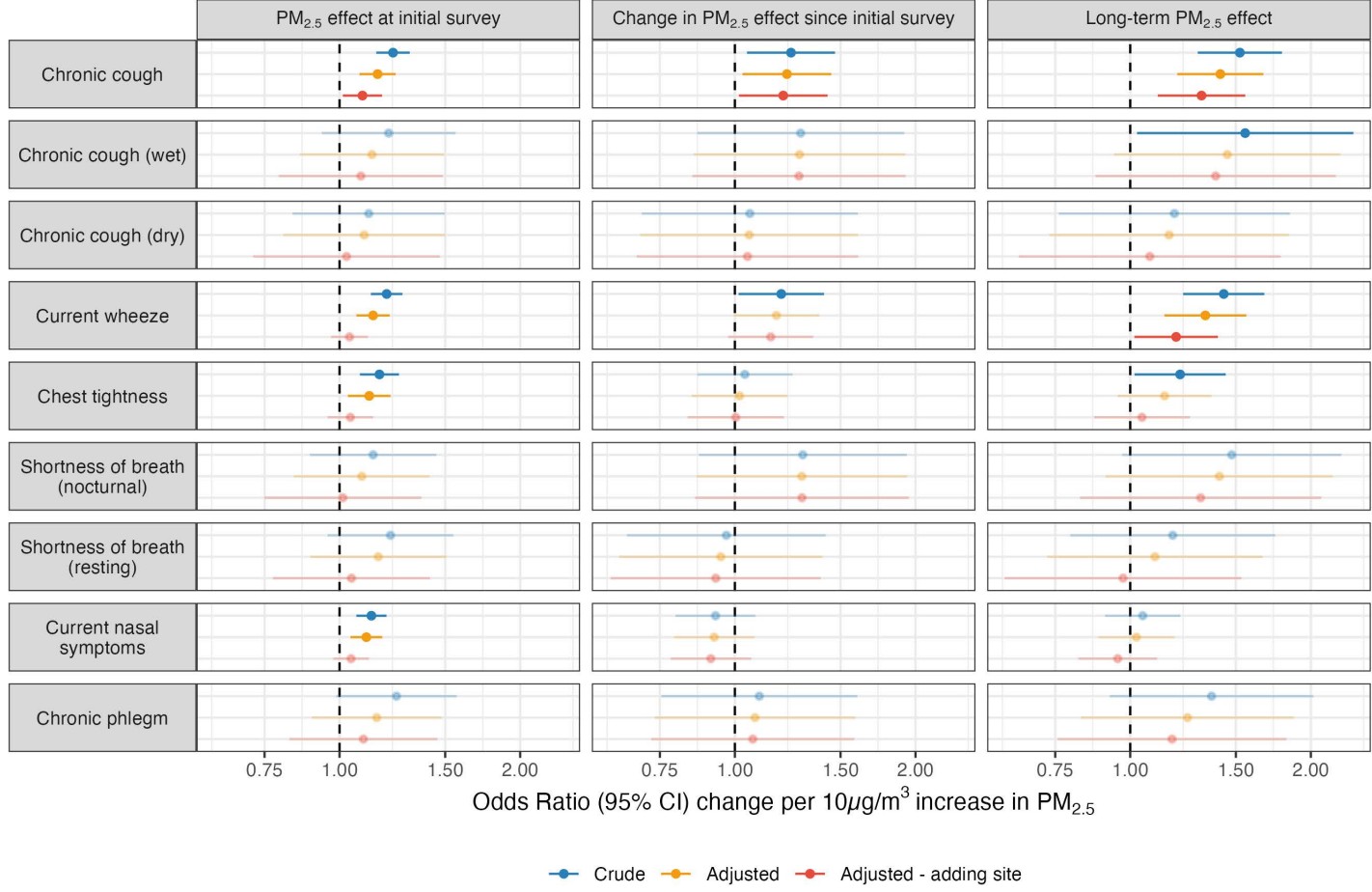

**Fig 2. Effects of 10 μg/m³ increase in daily mean coalmine fire-related PM$_{2.5}$ on risk of respiratory symptoms at the initial 2016/17 survey (2–3 years post-fire), change between the two survey rounds (interaction term between fire-related PM$_{2.5}$ and survey round), and the long-term effect at the 2022 follow-up (8.5–9 years post-fire, estimated using linear combination of PM$_{2.5}$ and interaction term); faded points and intervals indicate non-significant effects.**

attenuation to non-significance after inclusion of confounders and study site. The long-term effects of $PM_{2.5}$, or the association between $PM_{2.5}$ exposure and risk of respiratory symptom 8.5–9 years post-fire, were increased risk of both chronic cough and current wheeze. While effects among chronic cough subtypes were mostly non-significant, the direction and magnitude of associations indicated the increase in overall chronic cough over time was mostly attributable to wet cough.

To investigate whether $PM_{2.5}$ effects on current wheeze were the result of worsening asthma control, we stratified analyses based on self-reported asthma diagnoses at the initial 2016/17 survey. Perhaps surprisingly, effects were largely isolated to non-asthmatics. There was also a total reduction in chest tightness among asthmatics. These results are presented in Fig D in S1 Text.

### 3.3 Interaction effects of $PM_{2.5}$ and COVID-19 on respiratory symptoms

Model results from these analyses are summarised in Fig 3 and Table D in S1 Text. While there were no significant interactions between $PM_{2.5}$ exposure and COVID-19, point estimates were stable and positive for chronic wet cough (OR range across crude, adjusted,

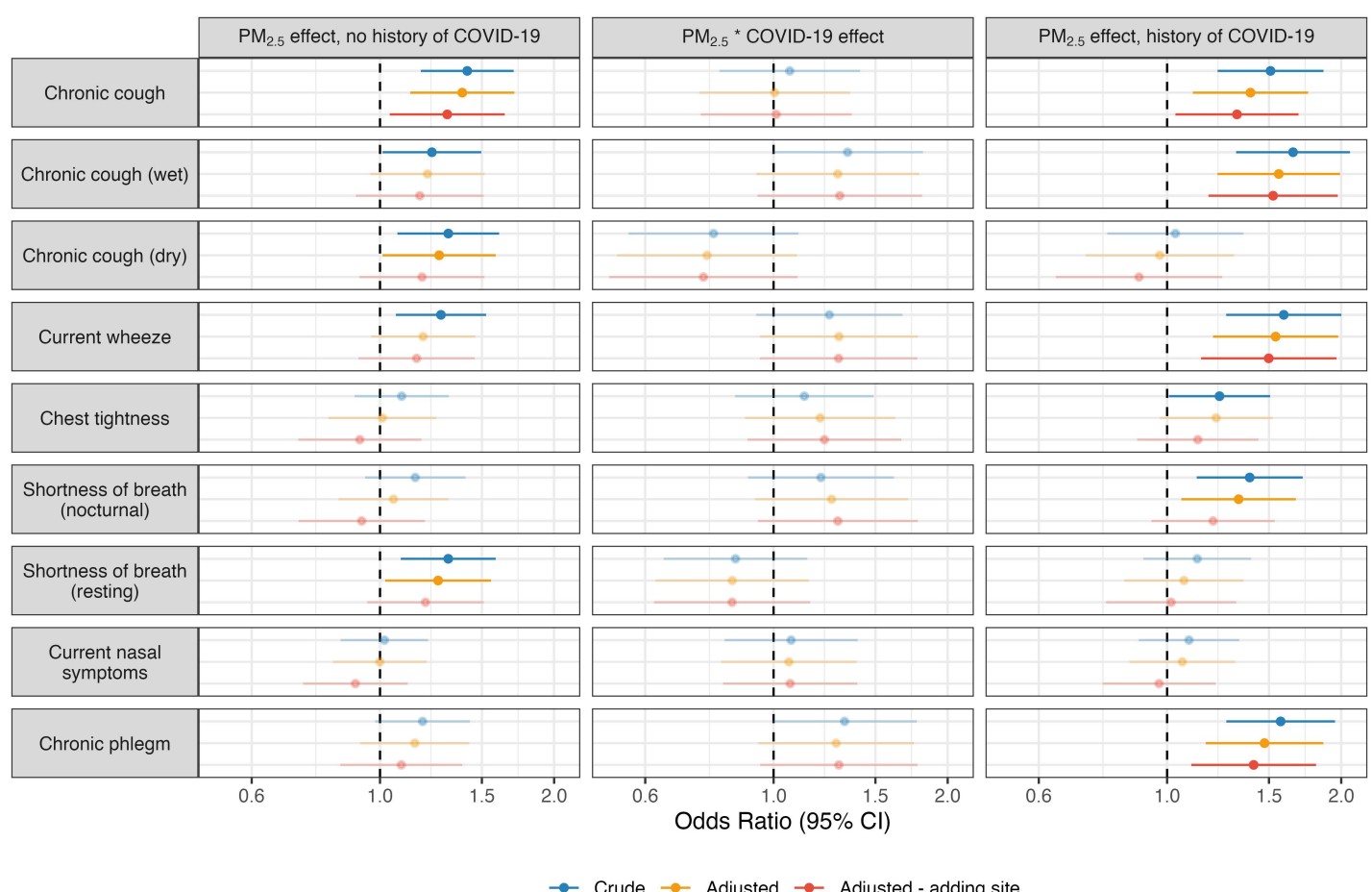

**Fig 3. Moderating effect of COVID-19 on relationship between daily mean coalmine fire-related $PM_{2.5}$ (10 µg/m³) and risk of respiratory symptoms at the 2022 follow-up survey (8.5–9 years post-fire, estimated using linear combination of $PM_{2.5}$ and interaction term); faded points and intervals indicate non-significant effects.**

and adjusted – adding site models: 1.29–1.34) current wheeze (OR range: 1.25–1.30), chest tightness (OR range: 1.13–1.22), nocturnal shortness of breath (OR range: 1.21–1.29), and chronic phlegm (OR range: 1.28–1.33), suggesting some exacerbation of $PM_{2.5}$ effects by COVID-19. Conversely, interactions trended negative with chronic dry cough (OR range: 0.76–0.79) and resting shortness of breath (OR range: 0.85–0.86). Chronic cough interactions approximated null (OR range: 1.00–1.07).

Notably, there appeared to be some variation in the effects of $PM_{2.5}$ exposure on risk of respiratory symptoms based on history of COVID-19. Among those reporting a history of COVID-19, $PM_{2.5}$ effects were larger and more consistently significant on the risk of chronic wet cough, current wheeze, nocturnal shortness of breath, and chronic phlegm, as well as possibly chest tightness. However, $PM_{2.5}$ effects on the risk of chronic dry cough and resting shortness of breath were only detectable among those without a history of COVID-19.

## 4  Discussion

Our first research question concerned the long-term effects of coalmine fire-related $PM_{2.5}$ on respiratory symptoms. We found that $PM_{2.5}$ exposure from the Hazelwood coalmine fire was associated with a long-term increased risk of chronic cough and current wheeze, and that the effects appeared to increase over time. Chronic cough is of particular concern as it associated with deteriorations in physical, mental and social health [37], and treatments are limited and complex [38]. Therefore, these findings are a worrying development not just for our cohort, but anyone exposed to extreme smoke events from landscape fires, which are increasing in frequency and duration due to climate change.

One possible mechanism to explain the increase in chronic cough is cough hypersensitivity, which lowers the threshold for a physiological cough response [39,40] and has previously been associated with environmental triggers such as $PM_{2.5}$ [39]. Such an effect has been demonstrated after other discrete but extreme air pollutant exposures from events like the 9/11 terrorist attacks in New York and earthquake rescue operations [39]. The precise mechanism is unclear, though there is evidence for neuropathology such as upregulation of sensory neuroreceptors (e.g., P2X3, TRPA1, and TRPV1) in response to air pollutants [39,41] including $PM_{2.5}$ [42]. Yet it remains unknown why the effects worsened over time. One potential exacerbator of the coalmine fire's effects is the 2019/2020 bushfire season, also known as the Black Summer, which was unprecedented in terms of intensity, duration, and spread, with a resultant smoke plume covering much of Australia for several months, including Morwell and Sale [43]. Those made vulnerable to cough hypersensitivity by the coalmine fire may have been triggered by the Black Sumer 5–6 years later.

The finding that fire-related $PM_{2.5}$ exposure was associated with elevated and, in some cases, increasing risk of respiratory symptoms up to eight years post-fire is largely consistent with previous Hazelwood Health Study findings, though there are some exceptions that warrant further consideration. For instance, earlier analyses found exposure to $PM_{2.5}$ from the mine fire was associated with poorer respiratory function [13] that later attenuated, possibly indicating some recovery [18]. However, respiratory symptoms and respiratory function are related but separate outcomes with different mechanisms. As noted above, increased chronic cough may be due to epigenetic changes that induce cough hypersensitivity, while poorer respiratory function may be due to changes to the peripheral airway system and obstruction, along with accelerated pulmonary ageing [13,18]. With this difference in mechanism may come varied trajectories. Another comparison point is health service use. While Hazelwood Health Study analyses have generally detected short and

medium-term increases in hospital admissions, emergency presentations, and ambulance attendances related to respiratory conditions [8,15,16], longer-term analyses only observed an increase in emergency presentations [17]. However, these sorts of health service uses are an imperfect measure of health as numerous factors influence whether someone seeks healthcare, which need not necessarily correspondence with respiratory symptoms. Further, the respiratory symptoms affected by fire-related $PM_{2.5}$ exposure are not those that typically result in hospital, emergency, or ambulance care.

Our second research question investigated whether COVID-19 exacerbated the effects of coalmine fire-related $PM_{2.5}$. We found some evidence of this. While interaction terms between $PM_{2.5}$ and COVID-19 were non-significant, point estimates among several symptoms were positive and robust to adjustment, and some $PM_{2.5}$-respiratory symptom effects (chronic wet cough, current wheeze, nocturnal shortness of breath, chronic phlegm, and possibly chest tightness) were more consistently detected among participants with a history of COVID-19. Exacerbation of $PM_{2.5}$ effects by COVID-19 may also explain why the risk of current wheeze worsened in association with $PM_{2.5}$.

Sensitivity analyses indicated that the increase in current wheeze was isolated to non-asthmatics, and there was a *reduction* in chest tightness among asthmatics. One explanation is diagnosed asthmatics have access to inhaled medications that enable them to control respiratory symptoms like wheeze, while non-asthmatics do not. There is some circumstantial evidence for this including an increase in dispensing of prescribed asthma medications such as rescue inhalers following the 2013 Oregon wildfire season [44]. To our knowledge, this is the first study to provide evidence that COVID-19 may exacerbate the effects of historical exposure to extreme but discrete smoke from a coalmine fire on risk of respiratory symptoms. However, given the tenuous nature of these associations, we highlight them with due caution. It is further worth noting that in each case, associations between $PM_{2.5}$ and respiratory symptoms among those with a history of COVID-19 attenuated with each adjustment, raising the possibility of residual confounding.

Interestingly, fire-related $PM_{2.5}$ exposure was associated with risk of chronic cough regardless of COVID-19 history. Though there was limited statistical power for analysis of wet and dry chronic cough subtypes, the patterns of association suggested that contracting COVID-19 increased the likelihood that chronic cough, which may have developed due to smoke exposure, was later accompanied by phlegm, as opposed to developing a new wet cough. These association patterns included: 1) $PM_{2.5}$*COVID-19 interaction effects trending positive for the effect on wet cough and negative for dry cough; 2) among those with a history of COVID-19, a clear positive effect of $PM_{2.5}$ on wet cough and null effect on dry cough, while both were positively associated with $PM_{2.5}$ exposure among those without a history of COVID-19; and 3) a null interaction between $PM_{2.5}$*COVID-19 on overall chronic cough.

## 4.1 Strengths and limitations

Among this study's strengths are the longitudinal design, use of time-location diaries and modelled air pollution data to estimate individual-level $PM_{2.5}$ exposure, accounting for events occurring between survey rounds like the COVID-19 pandemic, adjustment for important confounders such as socioeconomic factors with multiple measures, and repeated use of standardised validated measures for respiratory symptom outcomes.

However, there were several limitations. The participation rate and sample size in the follow-up survey were modest. Follow-up participants differed from the rest of the cohort on several key characteristics, which may have introduced selection bias. However, these factors were partially accounted for through adjustment in our regression models. In addition, we

lacked participant data on respiratory symptoms from before the coalmine fire. Time-location diaries that were used to determine where participants were during the mine fire probably resulted in some degree of measurement error, especially since they were administered 2–3 years after the event. Further, $PM_{2.5}$ exposure was based on modelled estimates, which could deviate from actual concentrations [4], and do not account for individual variations due to wearing masks or time spent outdoors. The uniqueness of this cohort, particularly their older age, regional location, and residence within a developed country with high vaccine uptake before COVID-19 became widespread, means our findings may not generalise findings to other settings.

## 5  Conclusions

Our findings suggest extreme but medium-duration $PM_{2.5}$ exposure from a coalmine fire has negative long-term effects on respiratory health, most noticeably chronic cough, possibly due to inducement of cough hypersensitivity, and current wheeze. This is concerning because the effects appeared to worsen over time, and chronic cough is associated with a deterioration in health. We also found some evidence that COVID-19 exacerbated effects of coalmine fire-related $PM_{2.5}$.

There are serious implications for the millions of Australians affected by the 2019/2020 Black Summer bushfires, the hundreds of millions affected by 2020 and 2023 wildfires across North America and elsewhere. As climate change increases the frequency, intensity, duration, and spread of fires, smoke exposure will grow as a global public health problem, and along with habitat destruction and the wild animal trade, may be exacerbated by the emergence of new zoonotic diseases.

## Supporting information

**S1 Text.  Supplementary materials.**
(DOCX)

## Author contributions

**Conceptualization:** Tyler J. Lane, Tracy A. McCaffrey, Jillian Ikin, Michael J. Abramson.

**Data curation:** David Brown, David Poland, Shantelle Allgood.

**Formal analysis:** Tyler J. Lane, Catherine L. Smith, Caroline X. Gao.

**Funding acquisition:** Jillian Ikin, Michael J. Abramson.

**Investigation:** Tyler J. Lane, Matthew Carroll, Amanda Johnson, Shantelle Allgood, Jillian Ikin, Michael J. Abramson.

**Methodology:** Tyler J. Lane, Matthew Carroll, Tracy A. McCaffrey, Michael J. Abramson.

**Project administration:** Tyler J. Lane, Matthew Carroll, David Brown, David Poland, Shantelle Allgood.

**Software:** David Brown.

**Supervision:** Michael J. Abramson.

**Visualization:** Tyler J. Lane.

**Writing – original draft:** Tyler J. Lane.

**Writing – review & editing:** Matthew Carroll, Brigitte M. Borg, Tracy A. McCaffrey, Catherine L. Smith, Caroline X. Gao, David Brown, Amanda Johnson, David Poland, Shantelle Allgood, Jillian Ikin, Michael J. Abramson.

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
