## [Decision Letter · Decision Letter 0]

20 Aug 2024

PGPH-D-24-00948

Respiratory symptoms after coalmine fire and pandemic: a longitudinal analysis of the Hazelwood Health Study adult cohort

Dear Dr. Lane,

Thank you for submitting your manuscript to PLOS Global Public Health. After careful consideration, we feel that it has merit but does not fully meet PLOS Global Public Health’s publication criteria as it currently stands. Therefore, we invite you to submit a revised version of the manuscript that addresses the points raised during the review process.

The authors need to explain how the measurement error in exposure assessment during the coalmine fire was minimized by combining self-reported time-location diaries with modelled PM2.5 estimates. Specifically, the spatial models to estimate individual PM2.5 exposure during the coalmine fire can have potential ecological bias. They can result in exposure misclassification due to variations in individual movement and duration of stay in the assessed areas.

Further, it is not clear if the outcome was assessed and excluded at baseline to follow-up only those at risk of developing respiratory symptoms. Also, the frequency of administering the question (whether weekly, monthly, or only towards the end) is not provided. It is unclear how the authors excluded prevalent symptoms at baseline. Since respiratory symptoms can be associated with other exposures, authors need to specify how the specificity and temporality are established between PM2.5 and the outcome.

The authors need to specify the hypothesis using a directed acyclic graph and justify the inclusion of confounders. For example, it is unclear how being fully vaccinated against COVID-19 can be a confounder given that it is not associated with PM 2.5 exposure.

Only 25% of the original cohort's 4,056 members participated in the follow-up survey. The authors need to explain why the differential loss to follow-up might have biased the study results. The discussion lacks details on potential threats to internal validity, such as differential loss to follow-up, measurement error, and the selection of confounders.

We look forward to receiving your revised manuscript.

Kind regards,

Giridhara R Babu, MBBS, MPH, PhD

Academic Editor

Journal Requirements:

1. When completing the data availability statement of the submission form, you indicated that you will make your data available on acceptance. We strongly recommend all authors decide on a data sharing plan before acceptance, as the process can be lengthy and hold up publication timelines. Please note that, though access restrictions are acceptable now, your entire data will need to be made freely accessible if your manuscript is accepted for publication. This policy applies to all data except where public deposition would breach compliance with the protocol approved by your research ethics board. If you are unable to adhere to our open data policy, please kindly revise your statement to explain your reasoning and we will seek the editor's input on an exemption. Please be assured that, once you have provided your new statement, the assessment of your exemption will not hold up the peer review process.

2. Figure 1: please (a) provide a direct link to the base layer of the map (i.e., the country or region border shape) and ensure this is also included in the figure legend; and (b) provide a link to the terms of use / license information for the base layer image or shapefile. We cannot publish proprietary or copyrighted maps (e.g. Google Maps, Mapquest) and the terms of use for your map base layer must be compatible with our CC-BY 4.0 license. 

* Natural Earth - All maps are public domain. (http://www.naturalearthdata.com/about/terms-of-use/ )

Additional Editor Comments (if provided):

The authors need to explain how the measurement error in exposure assessment during the coalmine fire was minimized by combining self-reported time-location diaries with modelled PM2.5 estimates. Specifically, the spatial models to estimate individual PM2.5 exposure during the coalmine fire can have potential ecological bias. They can result in exposure misclassification due to variations in individual movement and duration of stay in the assessed areas.

Further, it is not clear if the outcome was assessed and excluded at baseline to follow-up only those at risk of developing respiratory symptoms. Also, the frequency of administering the question (whether weekly, monthly, or only towards the end) is not provided. It is unclear how the authors excluded prevalent symptoms at baseline. Since respiratory symptoms can be associated with other exposures, authors need to specify how the specificity and temporality are established between PM2.5 and the outcome.

The authors need to specify the hypothesis using a directed acyclic graph and justify the inclusion of confounders. For example, it is unclear how being fully vaccinated against COVID-19 can be a confounder given that it is not associated with PM 2.5 exposure.

Only 25% of the original cohort's 4,056 members participated in the follow-up survey. The authors need to explain why the differential loss to follow-up might have biased the study results. The discussion lacks details on potential threats to internal validity, such as differential loss to follow-up, measurement error, and the selection of confounders.

Reviewers' comments:

Reviewer's Responses to Questions

**Comments to the Author**

1. Does this manuscript meet PLOS Global Public Health’s publication criteria ? Is the manuscript technically sound, and do the data support the conclusions? The manuscript must describe methodologically and ethically rigorous research with conclusions that are appropriately drawn based on the data presented.

Reviewer #1: Yes

Reviewer #2: Yes

Reviewer #3: Yes

2. Has the statistical analysis been performed appropriately and rigorously?

Reviewer #1: Yes

Reviewer #2: Yes

Reviewer #3: Yes

3. Have the authors made all data underlying the findings in their manuscript fully available (please refer to the Data Availability Statement at the start of the manuscript PDF file)?

Reviewer #1: No

Reviewer #2: No

Reviewer #3: Yes

4. Is the manuscript presented in an intelligible fashion and written in standard English?

Reviewer #1: Yes

Reviewer #2: Yes

Reviewer #3: Yes

5. Review Comments to the Author

Reviewer #1: It was an interesting paper on the long term impact of PM2.5 exposure on respiratory health.

Here are my specific comments

2. Methods

2.1.1

Exposures

Figure 1: What were the PM2.5 levels over the years (post 2014)?

2.2 Statistical analysis

Page No. 7

Each model adjusted for the presence of the specific respiratory symptom at the baseline survey (e.g., the model assessing whether COVID-19 exacerbated effects of PM2.5 exposure on prevalence of chronic cough at the 2022 follow-up survey adjusted for presence of chronic cough at the 2016/2017 baseline survey).

I think the word ‘was’ is missing in the sentence.

Each model was adjusted for the presence of the specific respiratory symptom at the baseline survey (e.g., the model assessing whether COVID-19 exacerbated effects of PM2.5 exposure on prevalence of chronic cough at the 2022 follow-up survey, was adjusted for presence of chronic cough at the 2016/2017 baseline survey).

3. Results

3.1 Descriptives

Page 9

National Death Index data (31), which were linked to cohort data after the survey was completed, indicated 143 invited cohort members (7.7%) died prior to the close of the survey window (December 2022)

Correct the percentage. It is 5.8% (143/2458)

4. Discussion

Could include comparison with studies on long term impact of wildfires on respiratory outcomes since there is a mention of the 2019/2020 Black Summer bushfires.

Increase in current wheeze was isolated to non-asthmatics, and there was a reduction in chest tightness among asthmatics. This is an interesting finding, could include supporting literature.

Reviewer #2: The study is highly relevant as it allows one to analyze the long-term effects of extreme “point-in-time” exposure to PM2.5. Such studies are not common, particularly when there is the possibility of using historical data to analyze long-term effects. The study also highlights the limited analysis of the potential cumulative effects between SARS-CoV-2 infection and respiratory symptoms. Specific suggestions for revision could enhance the quality of the study.

Abstract

The abstract does not clearly state the study's objective. I suggest including it, considering the research questions presented at the end of the introduction.

In the methods section, I suggest distinguishing between the models used to evaluate the long-term effects of PM2.5 exposure and the association of these effects with SARS-CoV-2 infection.

Introduction

-2nd Paragraph – In this second paragraph, what does GP mean?

- 2nd Paragraph – At the end of the paragraph, an increase in outcomes associated with the 2014 coalmine fire is described. Does this increase refer to other locations or other periods analyzed?

Methods

- Survey – It is not clear whether the profile of survey respondents changed compared to the 2016/2017 survey. It also does not mention how the survey was conducted, whether through personal interviews or using tools such as mobile apps. I suggest including this information.

- Exposure – I suggest including an explanation for choosing Sale as a control.

- Statistical Analyses – Regarding missing data, did this refer to the absence of some responses or respondents?

Results

- Regarding missing data, it is unclear how many imputations were made relative to the data obtained.

- To better understand the results, I strongly suggest including p-values in the text when significance is obtained and, if possible, the final models for each analysis in addition to Figures 2 and 3.

Discussion

- 1st Paragraph – At the end of the paragraph, it is mentioned that other locations worldwide also experienced an increase in chronic cough following exposure to extreme smoke events. I suggest referencing these other events and their consequences in exacerbating chronic cough.

- 2nd Paragraph - The 2019/2020 Black Summer bushfires are mentioned, but there is no further explanation of what this event was. Considering the publication's global reach, I suggest explaining what this refers to and including references, including regarding the prevalence of chronic cough.

- Finally, there is no mention of hospitalization data or records of care related to the respiratory symptoms analyzed. Although such data were not used in the proposed models, I suggest including them in the discussion or at least indicating the rationale for not using this data.

Reviewer #3: The longitudinal study provided the necessary data sufficient to address the research questions. The longitudinal study provides evidences of the effect of extreme exposure to PM_2.5 on the respiratory. The sample size isn’t large enough to capture the effect of PM_2.5 exposure, although this has been addressed as one of the limitation of the study. The method in the abstract did not address the method of analysis used in analyzing the data.

6. PLOS authors have the option to publish the peer review history of their article (what does this mean? ). If published, this will include your full peer review and any attached files.

**Do you want your identity to be public for this peer review?** For information about this choice, including consent withdrawal, please see our Privacy Policy .

Reviewer #1: No

Reviewer #2: No

Reviewer #3: **Yes: ** Adedoyin John-Joy Owolade

While revising your submission, please upload your figure files to the Preflight Analysis and Conversion Engine (PACE) digital diagnostic tool, https://pacev2.apexcovantage.com/ . PACE helps ensure that figures meet PLOS requirements. To use PACE, you must first register as a user. Registration is free. Then, login and navigate to the UPLOAD tab, where you will find detailed instructions on how to use the tool. If you encounter any issues or have any questions when using PACE, please email PLOS at figures@plos.org. Please note that Supporting Information files do not need this step.

---

## [Decision Letter · Decision Letter 1]

31 Dec 2024

Respiratory symptoms after coalmine fire and pandemic: a longitudinal analysis of the Hazelwood Health Study adult cohort

PGPH-D-24-00948R1

Dear Dr Lane,

We are pleased to inform you that your manuscript 'Respiratory symptoms after coalmine fire and pandemic: a longitudinal analysis of the Hazelwood Health Study adult cohort' has been provisionally accepted for publication in PLOS Global Public Health.

Best regards,

Giridhara R Babu, MBBS, MPH, PhD

Academic Editor

Reviewer Comments (if any, and for reference):

Reviewer's Responses to Questions

**Comments to the Author**

1. If the authors have adequately addressed your comments raised in a previous round of review and you feel that this manuscript is now acceptable for publication, you may indicate that here to bypass the “Comments to the Author” section, enter your conflict of interest statement in the “Confidential to Editor” section, and submit your "Accept" recommendation.

Reviewer #1: All comments have been addressed

2. Does this manuscript meet PLOS Global Public Health’s publication criteria ? Is the manuscript technically sound, and do the data support the conclusions? The manuscript must describe methodologically and ethically rigorous research with conclusions that are appropriately drawn based on the data presented.

Reviewer #1: Yes

3. Has the statistical analysis been performed appropriately and rigorously?

Reviewer #1: Yes

4. Have the authors made all data underlying the findings in their manuscript fully available (please refer to the Data Availability Statement at the start of the manuscript PDF file)?

Reviewer #1: No

5. Is the manuscript presented in an intelligible fashion and written in standard English?

Reviewer #1: Yes

6. Review Comments to the Author

Reviewer #1: None

7. PLOS authors have the option to publish the peer review history of their article (what does this mean? ). If published, this will include your full peer review and any attached files.

**Do you want your identity to be public for this peer review?** For information about this choice, including consent withdrawal, please see our Privacy Policy .

Reviewer #1: No
